# Unique Properties of Nutrient Channels on *Plasmodium*-Infected Erythrocytes

**DOI:** 10.3390/pathogens12101211

**Published:** 2023-10-02

**Authors:** Sanjay Arvind Desai

**Affiliations:** Laboratory of Malaria and Vector Research, National Institute of Allergy and Infectious Diseases, National Institutes of Health, Rockville, MD 20852, USA; sdesai@niaid.nih.gov

**Keywords:** malaria, nutrient uptake, ion channel, selectivity, conductance, permeation, antimalarial drug discovery

## Abstract

Intracellular malaria parasites activate an ion and organic solute channel on their host erythrocyte membrane to acquire a broad range of essential nutrients. This plasmodial surface anion channel (PSAC) facilitates the uptake of sugars, amino acids, purines, some vitamins, and organic cations, but remarkably, it must exclude the small Na^+^ ion to preserve infected erythrocyte osmotic stability in plasma. Although molecular, biochemical, and structural studies have provided fundamental mechanistic insights about PSAC and advanced potent inhibitors as exciting antimalarial leads, important questions remain about how nutrients and ions are transported. Here, I review PSAC’s unusual selectivity and conductance properties, which should guide future research into this important microbial ion channel.

## 1. Introduction

Malaria parasites are successful pathogens that grow and replicate within circulating erythrocytes of their vertebrate hosts. Growth within erythrocytes affords access to hemoglobin as an amino acid source and facilitates evasion of host immunity, but it also limits parasite access to nutritive solutes in host plasma. After invasion, the parasite remodels the host cell to access these nutrients and meet other requirements for intracellular survival. In the case of the human *P. falciparum* malaria parasite, this remodeling is achieved through the export of many proteins into the host cell to alter deformability, cytoadherence, and membrane permeability. 

The increased permeability of infected cells was originally identified through measurements of cation contents in blood from infected monkeys [1]. Studies over the next five decades used tracer flux and semi-quantitative osmotic fragility to tabulate the broad range of solutes with increased permeability and to identify several nonspecific inhibitors [2,3,4,5,6,7]. As these methods are limited to measurements on populations of cells, these studies were unable to distinguish between possible mechanisms, with proposals including fluid-phase endocytosis, a membrane-bound duct between the erythrocyte surface and the parasitophorous vacuole, nonspecific leaks, upregulated channels or transporters, and one or more parasite-associated transport proteins [8,9,10]. This uncertainty was resolved with cell-attached and whole-cell patch-clamp of individual cells, revealing a single ion channel, termed the plasmodial surface anion channel (PSAC) [11]. This channel was seen only on infected cells and was absent from co-cultured uninfected cells; through comparison of single-molecule currents and the total current from individual cells, the PSAC copy number was found to be 1000–2000 channels per trophozoite-infected cell. Subsequent patch-clamp studies from other groups suggested several upregulated human channels [12], adding to the research community’s interest in this important infected cell phenotype and raising questions about the number of infection-associated channels and whether they are host or parasite encoded. 

Some of these questions were subsequently addressed by chemical screens, which identified ISPA-28, an inhibitor that blocks uptake into cells infected with the Dd2 parasite line but not those infected with other clones [13]. This clone-dependent inhibition affected diverse organic solutes as well as Cl^−^ transport in a single-channel patch-clamp. In light of ISPA-28’s uniquely specific activity, this finding provided compelling evidence for PSAC as the primary mechanism for increased permeability of nearly all solutes after infection. Ca^++^, an essential divalent cation whose permeability is also increased after infection [14], appears to be the sole exception, as ISPA-28 and other PSAC inhibitors do not block its uptake [15,16]. 

Genetic mapping in a Dd2 × HB3 parasite cross and DNA transfection experiments identified two closely related parasite *clag3* genes as the sole determinant of the ISPA-28 block [13]. The encoded CLAG3 protein is trafficked to and inserted in the host membrane, consistent with the previously determined site of PSAC-mediated transport. Protease treatment of infected cells compromises PSAC activity, providing an additional transport phenotype that directly maps to *clag3* [17]. CLAG3 has been further implicated through molecular studies of PSAC mutants and growth inhibition studies [18,19]. A CLAG3 knockout parasite is viable with only partially compromised permeability, suggesting that *clag* paralogs on other chromosomes (*clag2*, *clag8*, and *clag9* in *P. falciparum*) also contribute to PSAC formation [20]. RhopH2 and RhopH3, encoded by single-copy genes in *P. falciparum*, associate with CLAG proteins, forming a stable complex that remains intact upon reaching the host membrane (termed the RhopH complex, [21,22,23,24,25]). 

Recently, a high-resolution structure of these proteins was solved using freeze–thaw release and purification of the soluble RhopH complex from *P. falciparum* culture [26]. This structure was subsequently confirmed using a novel cryoID algorithm that does not require protein purification [27,28]. Both structures represent the soluble RhopH complex, which is assembled during or shortly after translation of member proteins in mature schizont-infected cells [22,24]. This complex is trafficked via rhoptry organelles in the invasive merozoite to new erythrocytes upon invasion, where it eventually reaches and inserts in the erythrocyte membrane to form the channel. The structure of the membrane-embedded complex remains to be solved and would advance this field significantly. Can protein structure prediction using AlphaFold and the improved AlphaFold2 neural network-based algorithms help fill this gap? These algorithms have significantly advanced protein structure prediction and have even faithfully predicted challenging membrane protein structures [29,30]. Although my analyses using these algorithms revealed that AlphaFold yields an accurate prediction of the soluble RhopH complex’s structure, there are several reasons to be cautious about predicting the structure of the membrane-embedded PSAC complex. First, it is unclear whether the RhopH complex directly forms this channel, even though this prevailing model is supported by in vitro selection and DNA transfection studies, which have revealed mutations that govern inhibitor action, channel gating, and ion permeation [17,18,20,31]. Second, the unusual properties of this parasite channel and the lack of sequence homology to known channel proteins necessitate experimental determination of the structure. A similar view is held by structural biologists studying more well-characterized membrane proteins and drug targets [30,32].

Consistent with the model of channel formation directly by RhopH proteins, both PSAC activity and genes for each member of the RhopH complex are conserved in all examined members of the genus *Plasmodium*, implicating an essential role and a unique adaptation of malaria parasites to intracellular development within erythrocytes [33]. Remarkably, however, the RhopH proteins lack homology to known ion channels; they also lack the number of predicted transmembrane domains seen in most other ion channels.

Although the CLAG3 knockout line can be propagated at uncompromised rates in the standard nutrient-rich RPMI 1640-based medium, it fails to grow in a modified PGIM medium (PSAC growth inhibition medium) that has lower, more physiological concentrations of three nutrients with primary uptake via PSAC [18,20]. Genetic mapping of ISPA-28-mediated growth inhibition using PGIM and the Dd2 × HB3 genetic cross again implicated the *clag3* genes. All PSAC inhibitors, but not antimalarial drugs inactive against solute uptake, exhibit significantly improved parasite killing efficacies in PGIM [18]. These findings provided the first experimental evidence for an essential PSAC role in nutrient uptake for the intracellular parasite. Host cation remodeling and infected cell volume regulation are also primarily determined by PSAC-mediated ion transport and have been proposed as additional PSAC roles [34], but studies using sucrose-based media have excluded a requirement in parasite development [35]. 

While biochemical and molecular studies over the past two decades have provided fundamental mechanistic and physiological insights into PSAC and its essential role in nutrient uptake, many important questions remain. In addition to the much-debated question of whether the RhopH proteins directly form the PSAC nutrient uptake pore [36], an important future direction is to examine PSAC’s remarkable solute selectivity and permeation properties. Here, I describe the most unusual of these properties, which raise fundamental questions about how PSAC recognizes and transports solutes across the infected erythrocyte membrane. 

## 2. Na^+^ Exclusion despite Broad Permeability to Diverse Solutes

PSAC’s most unusual property may be its unparalleled ability to exclude Na^+^ while still transporting larger solutes of varying charge, shape, and size. Studies from numerous groups have tallied a diverse list of permeant solutes, including anions, monovalent cations, and organic solutes, with a partial tally in Table 1. For solutes whose transport rates have been quantitively measured, this table includes permeability estimates relative to inorganic Cl^−^, a physiologically relevant anion with high PSAC permeability (*P/P_Cl_*). Table 1 also includes measured ionic radii for inorganic anions and cations in angstroms; for organic solutes and ions without measured estimates, the van der Waals molecular radius, an alternate measure of the solute’s size in aqueous solution, was calculated using a validated approach [37].

Although Table 1 is not comprehensive, it shows the remarkably broad collection of solutes with known PSAC permeability. The high permeability to sorbitol forms the basis of synchronization of in vitro *P. falciparum* cultures and of percoll-sorbitol density gradient enrichment [45,46]. Permeability to toxins such as pepstatin, blasticidin, and leupeptin as well as NHS esters, all of which have relatively large molecular volumes, highlights how large solutes can pass through the channel pore and affect intracellular parasite development. In addition to high permeability to inorganic anions, the channel also has relatively high permeability to organic cations such as guanidinium^+^ and phenyl-trimethylammonium^+^, revealing that a solute’s net charge does not itself exclude permeation. 

Considering the broad range of solutes with measurable flux through PSAC, the stringent exclusion of Na^+^, whose permeability we estimate at ~10^−5^–10^−6^ relative to Cl^−^, is unprecedented when compared to other broad selectivity channels, which achieve no more than 30-fold exclusion of individual small ions (Table 2). The inability of these channels to exclude small ions makes sense: high transport rates along with promiscuous passage of solutes of various sizes and charges are difficult to achieve if any single ion must be excluded.

Although PSAC does not completely exclude Na^+^, its ability to keep this ion out compares favorably to Na^+^ exclusion by K^+^ channels, where *P_Na_/P_K_* is also thought to be in the 10^−4^–10^−6^ range [51]. Although the precise mechanism of Na^+^ exclusion despite very high K^+^ transport rates in K^+^ channels is still debated [52,53], an appealing model, sometimes referred to as the “snug fit” model, is that the dehydrated K^+^ is snuggly coordinated at one of four sequential binding pockets along its path through the channel pore [54]. The smaller Na^+^ ion, with a 0.95 Å radius in contrast to a K^+^ radius of 1.3 Å (Table 1), is not well-coordinated by carbonyl oxygens along the channel’s selectivity filter. In this model, K^+^ channels then achieve specific recognition of K^+^ and high transport rates by using multiple sequential binding sites along the pore that promote repulsion between K^+^ ions, leading to ions knocking each other through the channel. Na^+^ exclusion by K^+^ channels is essential for action potential generation in excitable cells, stimulating decades-long research into possible mechanisms by many accomplished laboratories. 

While Na^+^ exclusion by K^+^ channels is remarkable and its study has provided foundational insights into how channels work, I submit that PSAC’s ability to exclude Na^+^ with comparable efficiency is equally fundamental and may be even more puzzling: PSAC must allow many different solutes of varying size and charge to permeate and cannot utilize either “snug fit” for diverse solutes or the knock-through method available to K^+^ channels, which are tasked with only passing a single ion specifically.

Why did PSAC evolve to have such a remarkably low Na^+^ permeability? The leading hypothesis now is that Na^+^ exclusion is required to preserve infected erythrocyte osmotic stability in plasma [43], where Na^+^ and Cl^−^ represent the most abundant cation and anion. Because erythrocytes have a high Cl^−^ permeability, an increased PSAC permeability to Na^+^ would lead to net Na^+^Cl^−^ uptake because of their high concentrations in plasma. This uptake would be accompanied by water uptake through host-encoded aquaporins [55], leading to cell swelling and osmotic lysis [38]. Without the observed level of Na^+^ exclusion, *P. falciparum*-infected cells would lyse before the intracellular parasite is able to complete its lifecycle, leading to premature release from the cell and parasite death. Experimental evidence for this model comes from in vitro cultivation of *P. falciparum* in modified media that do not have high Na^+^ concentrations [35]. This study found that replacement of Na^+^ in the standard RPMI 1640-based culture medium with the corresponding K^+^ salts failed to support growth for even a single parasite cycle because of osmotic lysis. Remarkably, however, simple supplementation of this K^+^-based medium with 50 mM sucrose, an impermeant disaccharide that the parasite cannot utilize, restored continuous parasite growth and culture expansion. Donnan considerations indicate that 50 mM sucrose is sufficient to prevent osmotic lysis. Because K^+^ has a higher PSAC permeability than Na^+^ and sucrose has negligible PSAC permeability (Table 1), these observations strongly suggest that evolution selected for PSAC’s low Na^+^ permeability. If the channel’s Na^+^ permeability were modestly higher (e.g., similar to that of K^+^), infected cells would lyse in plasma as they do in the K^+^-based medium.

Evolutionary pressures also likely contributed to a K^+^ permeability greater than that of Na^+^. Uninfected erythrocytes maintain an outward electrochemical gradient for K^+^, so a comparatively higher K^+^ permeability will cause greater K^+^ efflux than Na^+^ entry during parasite maturation when PSAC activity gradually appears on the host membrane. This imbalance will, at least initially, lead to modest cell shrinkage and help avoid the osmotic lysis of infected cells.

Finally, while Na^+^ and K^+^ permeabilities in PSAC are surprisingly low, they are not zero. The slow but nonzero flux of these cations leads to a dramatic remodeling of cation concentrations in the erythrocyte cytosol as the parasite matures [1,34,35,56]: The normally low erythrocyte Na^+^ concentration increases from ~10 mM to >100 mM, while K^+^ decreases from ~120 mM to <20 mM. While several studies have suggested that these changes benefit the intracellular parasite by providing a host cytosolic composition that resembles the extracellular milieu, unabated parasite cultivation in sucrose-based media that fully prevents this remodeling excludes such a requirement [35]. These sucrose-based media also exclude the role of nonzero Na^+^ and K^+^ permeabilities in parasite egress as proposed [34]: Infected cells complete their intracellular cycle on an unchanged schedule in sucrose-containing media. Instead of egress through timed osmotic lysis, *P. falciparum* utilizes proteases and kinases to disrupt the host membrane and complete its intracellular parasite cycle [57,58]. 

## 3. Small Single Channel Conductance despite Permeability to Bulky Solutes

A second remarkable property is PSAC’s remarkably low ion and solute throughput despite its capacity to transport large organic solutes. The net transmembrane flow of a specific solute is the sum of flow through all relevant channels and transporters on a membrane. Because a large number of channels with low individual throughput can produce the same total flux as a small number with high transport rates, PSAC’s low throughput can only be appreciated by measuring transport through individual channel molecules. Fortunately, this can be precisely quantified with patch-clamp methods, which are able to track the picoampere currents associated with ion flow through single channel molecules in real time [59,60]. 

In these patch-clamp experiments, throughput is quantified as conductance (*γ*), which normalizes the measured single channel current (*i*) to the applied membrane potential (V_m_) according to *γ* = *i*/V_m_. Using this formalism, the first measurements of PSAC single channel currents revealed a remarkably low 20 picosiemen conductance, written as 20 pS or 0.02 nS. Table 3 compares this value to a few other channels that also transport large organic solutes. Conductance increases with the ion’s concentration at both faces of the channel, at least up to the concentrations that saturate the channel pore (Figure 1). Thus, comparisons of single-channel conductances should be made using similar recording solutions, as also listed in Table 3. 

For relatively nonselective channels such as those listed in this table, it is possible to predict the conductance’s value based on the simplifying assumption that broadly selective channels behave as rigid pores capable of passing any solute that can fit through the narrowest part of the pore. Such a channel’s conductance can then be calculated using Ohm’s law and geometric considerations according to *γ* = 1/*R_channel_*, where the resistance *R_channel_* is given by:(1)$$R_{channel}=\left(l+\frac{\pi r}{2}\right)*\frac{\rho }{\pi r^{2}}$$

In this equation, *l*, *r,* and ρ represent the pore length, pore radius, and recording solution resistivity, respectively (Figure 1a) [61]. The pore length, *l*, is adequately approximated by the lipid bilayer’s thickness because the aqueous conduit must fully traverse the membrane. The pore radius for broad selectivity channels can also be determined by finding the largest permeant solute through so-called solute exclusion experiments, as have been carried out for the channels listed in Table 3. Using Equation (1) and the results of these exclusion experiments, I calculated the predicted conductance for each of these channels. For these other channels, it is apparent that simple geometric considerations and exclusion experiments yield predicted values in good agreement with values measured from single-channel patch-clamp (within 3-fold of each other, Table 3). 

In contrast to these other channels, this calculation reveals that PSAC’s single channel conductance or ion throughput is nearly 100-fold lower than predicted based on the size of the largest permeant solute. The remarkably small currents tell us that ions flow through the open channel at a much slower rate than expected. Indeed, these small currents account for why molar salt solutions are required for single PSAC recordings in infected erythrocyte patch-clamp. This low flux does not result from saturation in the channel pore because measurements at a range of [Cl^−^] concentrations, including physiological levels, reveal a *K*_0.5_ for Cl^−^ of 1.05 M. The reasons for this lower-than-expected flux are presently unknown. Possible explanations include the binding of permeating solutes within the pore, interactions or repulsion between ions and solutes or water during transit, or a requirement for conformational changes in the channel protein during transport. These and other explanations deserve rigorous examination.

## 4. Higher Permeability of Large Ions

Another unexpected property of PSAC is apparent in Table 1: For both the halide anions and group 1A cations, permeability increases without exception as the ion gets larger. For anions, a rank order of SCN^−^ > I^−^ > Br^−^ > Cl^−^ was determined using reversal potential measurements with whole-cell patch-clamp, yielding thermodynamically precise permeability estimates under zero-flow conditions. For group 1A cations, Table 1 shows Cs^+^ > Rb^+^ > K^+^ >> Na^+^ with the bulky guanidinium cation (CH_6_N_3_^+^) some 80-fold more permeant than the large Cs^+^ cation. This observed greater flux of larger ions is, at first consideration, counterintuitive because geometric considerations suggest that small ions should more easily navigate a pore of finite radius (Figure 1b). After all, it is easier to throw a tennis ball than a basketball through a hoop!

This inverse permeability–size relationship has been observed in many selective cation and anion channels, where it has been reconciled by a requirement for ion dehydration before passage through the pore [65,66]. Because ions in bulk aqueous solution are surrounded by a shell of water molecules, energy must be expended for removal of these waters if the channel only passes ions without bound water. This energy may be partially or wholly compensated by a binding pocket within the pore (Figure 1c), as observed in K^+^ channels where carbonyl side chains stabilize the dehydrated K^+^ and offset the loss in Gibbs free energy upon dehydration [54]. Then, a channel that requires ion dehydration but provides minimal or no stabilization of the ion within the pore will favor ions that are most easily dehydrated, a combination sometimes referred to as a “weak binding site” [65]. Because monovalent ions with larger ionic radii distribute their single charge over a larger volume, the change in Gibbs free energy upon dehydration is typically inversely related to ion size. The significantly higher permeability of guanidinium^+^ and SCN^−^ than the next largest ion in each of these series further supports the hypothesis that permeation through PSAC requires ion dehydration because these ions distribute their single charges over larger effective areas [67], further reducing water binding and the associated Gibbs free energy. Thus, these permeability data strongly suggest that ions must be dehydrated to pass through PSAC and that the pore offers little shielding of the naked ion’s charge during permeation. 

While selective ion channels are sometimes known to exhibit inverse size-permeability relationships for cations or anions [68,69], this phenomenon is unusual for PSAC for two reasons. Most importantly, the radius of a hydrated Cl^−^ ion, 3.32 Å [70], is still less than the radius of sorbitol, a sugar alcohol with high PSAC permeability. So, we are left to wonder how and why PSAC requires dehydration of these monovalent ions before transport if the pore is large enough to accommodate larger organic solutes. Second, although channels selective for cations or anions are known to require ion dehydration, I could not find clear examples of other channels that pass cations, anions, and bulky organic solutes and yet require the removal of bound water molecules. 

Dehydration of ions as a requisite for transport raises the question of whether PSAC allows water to permeate. This question has not been examined in any published study to date, in large part because the high intrinsic water permeability of erythrocytes makes measuring possible increases after *Plasmodium* infection difficult. Indeed, the H_2_O permeability of human erythrocytes, estimated at ~2.5 × 10^−3^ cm/s [71], is some 10^4^ greater than the sorbitol permeability coefficient associated with PSAC. While it remains possible that PSAC does increase water permeability at the erythrocyte membrane, its effect on erythrocyte osmotic balance is likely to be negligible. 

While the mechanistic basis of this inverse size-permeability relationship in PSAC remains unclear, it seems likely that it evolved to facilitate Na^+^ exclusion: The Gibbs free energy required to dehydrate Na^+^, 365 kJ/mol [72], is among the highest of abundant ions in plasma. PSAC’s weak binding site offers little compensation for this high loss in free energy, rendering it very hard for Na^+^ to get through the pore. As discussed above, Na^+^ exclusion by PSAC is essential for osmotic stability of the infected erythrocyte in plasma.

## 5. Conclusions

Increased permeability to nutrients and many other solutes is one of the earliest cellular phenotypes identified for *Plasmodium*-infected erythrocytes [1]. While the past two decades have yielded fundamental insights into the ion channel mechanism and molecular basis of these increases, fundamental questions remain about how PSAC distinguishes between solutes to allow uptake of nutrients needed by the intracellular parasite. Most notably, how the channel excludes the small Na^+^ ion, a feat not matched in other broad selectivity channels and yet required for parasite survival, remains mysterious. While quantitative methods for studying transport through this channel have permitted measurements that range from single ion channel currents to high-throughput screens for chemical inhibitors [60,73], technological advances such as functional reconstitution of ion channels or heterologous expression of PSAC activity may be required to understand Na^+^ exclusion, the remarkably small single channel conductance, and the need for ion dehydration despite a large apparent pore diameter. A better understanding of these and other unusual features of this channel will provide insights into permeation through other microbial ion channels and should guide the development of improved PSAC inhibitors as candidate antimalarial drugs. 

## Figures and Tables

**Figure 1 pathogens-12-01211-f001:**
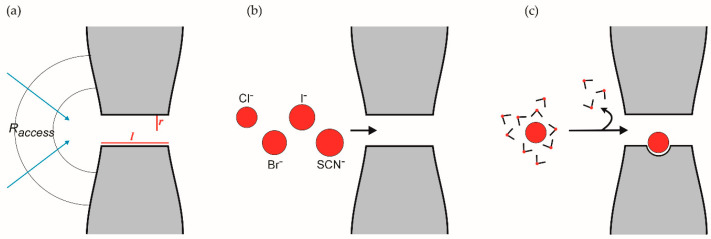
Permeation through an ion channel pore. (**a**) Geometric determinants of flow through a rigid channel pore [61]. A channel’s resistance to ion flow is given by *R_channel_* = *R_pore_* + *R_access_*. *R_pore_* is determined by the aqueous solution resistivity, ρ, and the pore’s radius and length, *r* and *l*, respectively, as $\frac{\rho *l}{\pi r^{2}}.$
*R_access_* is the resistance an ion encounters as it moves from bulk solution to the mouth of the pore and is given by $\frac{\rho }{2r}.$ Rearrangement of the sum of these terms yields Equation (1). (**b**) Schematic showing PSAC’s inverse size-permeability relationship with the largest anion, SCN^−^ having the greatest permeability. (**c**) Dehydration-dependent permeation through PSAC. The ion, shown surrounded by water molecules in bulk solution, is stripped of water before entry and stabilization at a binding site in the pore.

**Table 1 pathogens-12-01211-t001:** Relative PSAC permeabilities and dissolved radii of key solutes.

Solute	*P/P_Cl_*	Dissolved Radius (Å)	References
Halide/pseudohalide anions			
Cl^−^	1.0	1.67 *	[11]
SCN^−^	7.4	2.15–2.20 *	[11]
I^−^	3.5	2.06 *	[11]
Br^−^	1.2	1.95 *	[11]
Sugars and sugar alcohols			
sorbitol	0.008	3.4	[38]
Ribose	0.023	3.19	[39]
Glucose	0.005	3.38	[39]
Sedoheptulose	0.002	3.56	[39]
Sucrose	<<10^−4^	4.10	[39]
Organic anions and amino acids			
Acetate^−^	0.87	2.38	[11]
Lactate^−^	0.43	2.70	[11]
Glutamate^−^	0.11	3.18	[11]
Proline	0.018	2.96	[40]
Organic solutes and toxins			
Pepstatin	nonzero	5.55	[4]
Blasticidin	nonzero	4.45	[41]
Leupeptin	nonzero	4.73	[42]
Sulfo-NHS-LC-LC-biotin	nonzero	5.19	[43]
C_57_H_82_F_3_N_10_O_26_S_4_ (NHS ester)	nonzero	6.74	Unpublished
Organic and inorganic cations			
Guanidinium^+^	0.08	2.78 *	[44]
Phenyl-trimethylammonium^+^	0.02	3.26	[3,44]
Cs^+^	~10^−3^	1.9 *	[34,44]
Rb^+^	~10^−3^	1.7 *	[34,44]
K^+^	10^−3^–10^−4^	1.3 *	[34,44]
Na^+^	10^−4^–10^−6^	0.95 *	[34,44]

Permeant solutes whose *P/P_Cl_* could not be calculated are listed as having nonzero permeability. * Values represent ionic radii for these anions and cations; calculated van der Waals volume is listed for all other solutes.

**Table 2 pathogens-12-01211-t002:** Ion exclusion in other broad selectivity channels.

Channel	*P_cation_/P_Cl_*	References
VDAC	0.5	[47]
Bacterial porin	2–3.5	[48]
*P. falciparum* PVM channel	1.0	[49]
Maxi-anion channel	3.5	[50]

**Table 3 pathogens-12-01211-t003:** Comparison of measured and predicted conductances for broad selectivity channels.

Channel	*r*, nm	Recording Solution	*γ*, nS Measured	Reference	*γ*, nS Predicted Equation (1) *
Bacterial porin	0.7	1 M KCl	1.4	[62]	2.0
VDAC	2.0	1 M KCl	4.0	[47]	13.0
Phage export channel	3.0	150 mM KCl	3.4	[63]	3.1
*P. falciparum* PVM channel	1.15	100 mM KCl	0.155	[64]	0.52
PSAC	0.674 *	1.145 M Choline-Cl	0.02	Table 1; [11]	1.95

* For all channels, calculations used *l* = 7 nm. ρ, 90, 77, and 9.5 ohm*cm for 100 mM KCl, 150 mM KCl, and 1–1.145 M KCl or Choline-Cl. PSAC, *r* taken as the largest permeant solute in Table 1.

## Data Availability

The data presented are contained within the article tables.

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
