# Peer review of "Unique Properties of Nutrient Channels on Plasmodium-Infected Erythrocytes"

_pathogens, 2023, doi:10.3390/pathogens12101211_

Round 1

Reviewer 1 Report

In this manuscript, Desai reviews the current knowledge and main unanswered questions of the Plasmodium-encoded channel PSAC, one of the main actors of the increasing permeability of the erythrocyte’s membrane after Plasmodium infection. The author reviews the biochemical, structural and functional data supporting the identity of the protein components that compose the channel. Further, he discusses the physiopathological meaning of the unusual low Na+ specificity of PSAC despite its apparent tolerance to accept a wide range of both, charged and non-charged solutes. Next, data from single-channel recordings allows the author to discuss about the mechanistic model of solute permeation through PSAC, where selectivity and low conductance rates are dictated by the capacity of the channel to coordinate the non-hydrated solute and the energy penalty due to solute de-hydration before entering the channel.

Overall, the manuscript is highly comprehensive and well-written. Nevertheless, I have a question:

1.     Since maintaining the osmotic pressure after erythrocyte invasion seems to be one of the main and experimentally-supported hypothesis of PSAC’s biological role, I am wonder why the author did not discuss about any evidence (or lack of) of water permeation through PSAC .

Minor point:

For better reading, in figure 1 legend (line 217), please define the “recording solution density” term from the equation. Although, the author defines this term later in the text, it should be also discribed in the legend.

Reviewer 2 Report

This is a review on a channel in the host erythrocyte membrane activated by the malaria parasite. Overall, the article is well written and contributes to the field. 

One suggestion is to make a section on the identification of the channel protein and its structure, perhaps starting around line 44.

Related to the structure, has there been any modeling studies carried out on clag3, rhoph2, or rhoph3? Considering the ease at doing modeling studies, it might be worthwhile to include some models if something interesting is discovered. 

minor suggestions: 

line 8: delete ion since the channel is not exclusive to ions

line 24: delete cytosol. Many exported proteins are associated with membranes and cytosol is not the same as cytoplasm. 

line 43: change human to host

Table 1: not clear what non-zero means

line 223: Table 3?
